# Modularity Implications of an Overground Exoskeleton on Plantar Pressures, Strength, and Spasticity in Persons with Acquired Brain Injury

**DOI:** 10.3390/s24051435

**Published:** 2024-02-23

**Authors:** Carlos Cumplido-Trasmonte, Eva Barquín-Santos, María Dolores Gor-García-Fogeda, Alberto Plaza-Flores, David García-Varela, Leticia Ibáñez-Herrán, Carlos Alted-González, Paola Díaz-Valles, Cristina López-Pascua, Arantxa Castrillo-Calvillo, Francisco Molina-Rueda, Roemi Fernández, Elena García-Armada

**Affiliations:** 1International Doctoral School, Rey Juan Carlos University, 28922 Madrid, Spain; carlos.cumplido@marsibionics.com; 2Marsi Bionics SL, 28521 Madrid, Spain; eva.barquin@marsibionics.com (E.B.-S.); mariadolores.gor@urjc.es (M.D.G.-G.-F.); alberto.plaza@marsibionics.com (A.P.-F.); david.garcia@marsibionics.com (D.G.-V.); leticia.ibanez@marsibionics.com (L.I.-H.); elena.garcia@marsibionics.com (E.G.-A.); 3Department of Physical Therapy, Occupational Therapy, Rehabilitation and Physical Medicine, Faculty of Health Sciences, Rey Juan Carlos University, 28922 Madrid, Spain; francisco.molina@urjc.es; 4Spanish National Reference Centre for Brain Injury (CEADAC), 28034 Madrid, Spain; cgonzaalted@imserso.es (C.A.-G.); paoladiazvalles@gmail.com (P.D.-V.); 5Centro Lescer, 28035 Madrid, Spain; clopez@lescer.es (C.L.-P.); acastrillo@centrolescer.org (A.C.-C.); 6Centre for Automation and Robotics (CAR), CSIC-UPM, Ctra. Campo Real km 0.2–La Poveda-Arganda del Rey, 28500 Madrid, Spain

**Keywords:** acquired brain injury, modular, exoskeleton, plantar pressures, rehabilitation

## Abstract

This study explored the effects of a modular overground exoskeleton on plantar pressure distribution in healthy individuals and individuals with Acquired Brain Injury (ABI). The research involved 21 participants, including ABI patients and healthy controls, who used a unique exoskeleton with adaptable modular configurations. The primary objective was to assess how these configurations, along with factors such as muscle strength and spasticity, influenced plantar pressure distribution. The results revealed significant differences in plantar pressures among participants, strongly influenced by the exoskeleton’s modularity. Notably, significant distinctions were found between ABI patients and healthy individuals. Configurations with two modules led to increased pressure in the heel and central metatarsus regions, whereas configurations with four modules exhibited higher pressures in the metatarsus and hallux regions. Future research should focus on refining and customizing rehabilitation technologies to meet the diverse needs of ABI patients, enhancing their potential for functional recovery.

## 1. Introduction

Acquired Brain Injury (ABI) encompasses a spectrum of neurological impairments resulting from non-congenital factors, such as trauma, infection, or vascular events [1]. With an incidence that spans across diverse age groups, ABI presents a significant public health concern globally, affecting people worldwide and being the leading cause of ambulation deficits [2]. 

The restoration of walking is the goal most often stated by ABI patients and it is identified as one of the most important rehabilitation goals in ABI care [3]. Gait training plays a critical role in achieving this objective, focusing on improving the ability to walk independently and efficiently. Through targeted interventions, such as physical therapy, gait training aims to address specific impairments related to mobility, balance, and coordination that may arise following an ABI [4]. Effective gait rehabilitation not only contributes to the restoration of functional mobility but also enhances overall quality of life for individuals recovering from ABI [4].

In recent years, advancements in electromechanical technology, accelerated data processing, and reductions in equipment size have propelled exoskeletons into the forefront of rehabilitation for individuals with ABI [5]. These technological strides not only present a viable alternative to traditional gait rehabilitation methods, but also signify a paradigm shift [6]. In comparison to conventional therapies, robotic gait rehabilitation provides a highly controlled, repetitive, and intensive training regimen set within an engaging environment [7]. This not only alleviates the physical strain on therapists but also enables the delivery of objective and quantitative assessments of a patient’s progress. A recent Cochrane review revealed that patients who received robotic-assisted gait training in combination with physiotherapy after stroke were more likely to achieve independent walking than patients receiving gait training without such devices [8].

Within overground walking exoskeletons [9], a novel modularity concept is being implemented in recent prototype designs [10,11,12]. This design configures an exoskeleton with separate modular joints, which can be combined based on the patient’s degree of function. These systems offer flexibility in assembly, allowing specific configurations to address individual patients’ gait deficits. Therapists can select and arrange modules based on each patient’s functional capacity. This customization enables a tailored approach, optimizing the efficacy of the exoskeleton for individual user needs. Consequently, the modularity concept challenges the prevailing norms in current exoskeleton design [6].

The ankle and foot play a fundamental role in supporting the body, and any abnormal movement in these structures can directly impact the balance of the legs and trunk, leading to alterations in the overall gait pattern [13]. There is uncertainty regarding whether robotic exoskeletons for assisted walking, guided by normal and symmetrical kinematic trajectories of the hip and knee, also result in symmetrical and physiological functions of the foot, not influenced by the exoskeleton itself [14].

Plantar pressures in various regions of the foot have significant implications for individuals with ABI, affecting mobility and the risk of foot-related complications. High plantar pressures, especially in the central metatarsus, hallux, and medial metatarsus, are associated with overuse injuries in these areas, highlighting the need for targeted interventions to prevent or manage such injuries in patients with ABI [15] The use of high-heeled footwear shifts pressure and stress from the heel to the metatarsus and hallux, increasing the risk of deformities and pain, suggesting that footwear choice is crucial for managing foot health in ABI patients [16]. Additionally, biomechanical alterations, such as those caused by hallux valgus, modify plantar pressure distribution, increasing the load on the lateral metatarsus while reducing pressure under the hallux, potentially contributing to the development of metatarsalgia [17]. These changes in plantar pressure distribution may have significant clinical implications, including the development of pain, deformities, and decreased foot functionality, emphasizing the importance of preventive and therapeutic interventions aimed at improving plantar pressure distribution in at-risk individuals, such as those with ABI. To delve further into understanding how this novel modularity concept affects the gait of individuals with ABI, a comprehensive evaluation of its impact is essential. Consequently, the primary objective of this study is to compare the distinct influence of modularity on plantar pressures during walking with an exoskeleton in both healthy individuals and those with ABI. Additionally, the investigation aims to scrutinize the relations of module configuration selection with muscle strength and spasticity of lower limbs in individuals affected by ABI. Moreover, to date no studies have been found that provide information on plantar pressure values during walking with an overground exoskeleton. This novel aspect adds a unique contribution to the existing body of knowledge in the field.

## 2. Materials and Methods

This intervention trial was an open-label design within two groups and took place in three locations: at the Centre for Automation and Robotics (Spanish National Research Council, Madrid, Spain), in the Spanish Reference Centre for Brain Injury (CEADAC, Madrid, Spain), and in Centro Lescer (Madrid, Spain). Adherence to the Standard Protocol Items: Recommendations of Interventional Trials (SPIRIT) guidelines were ensured to maintain the study’s quality [18]. The research adhered to the ethical principles outlined in the Declaration of Helsinki [19]. Approval was obtained from the University Hospital of Getafe and the Spanish Drug and Medical Devices Agency (reference 961/21/EC R) before the initiation of the study. The clinical trial, registered on ClinicalTrials.gov (accessed on 1 November 2023) under NCT05265377, occurred in March and April of 2023.

### 2.1. Participants

This study focused on individuals diagnosed with ABI. In order to be eligible, participants had to meet specific inclusion criteria: (1) age within the range of 18 to 85 years, (2) weight below 100 kg, (3) height between 150 and 190 cm, (4) hip width ranging from 30 to 45 cm, (5) distance from the hip joint center to the knee joint center between 36 and 50 cm, (6) distance from the knee joint center to the ground between 43.5 and 59.5 cm, (7) European shoe size between 36 and 45, (8) capability to understand and follow simple commands, and (9) a Functional Ambulation Category (FAC) score equal to or below 4 [20].

Exclusion criteria for participation in the study included the following: (1) a lower limb spasticity score of 4 on the Modified Ashworth Scale (MAS) [21], (2) skin alterations in areas of contact with the device, (3) planned surgical intervention during the study, (4) two or more osteoporotic fractures in the lower limbs within the past 2 years, (5) exercise intolerance, (6) limb and/or spine surgery within the 3 months preceding the start of the study, and (7) psychiatric disorders that may interfere with proper device use or study participation, such as impulsivity or inability to understand simple commands.

The eligibility criteria for healthy participants were identical except for ABI diagnosis and the FAC score.

### 2.2. Device

STELO represents a functional modular overground exoskeleton (Figure 1). It enhances users’ motor capabilities by augmenting lower limb strength and mobility. A key feature is its functional modularity, enabling therapists to choose actuated joints based on patients’ functional needs. This selection is achieved by mechanically and electronically connecting/disconnecting the actuation modules. The decentralized control architecture eliminates the need for a central controller, utilizing local controllers for each module, corresponding to the hip and knee joints in the sagittal plane.

The STELO exoskeleton employs a scaled-down version of ARES technology [22] for actuation. This technology enables the adjustment of actuator stiffness and measures interaction forces in elastic elements connected to the actuator output, ensuring smoother power transmission, making it suitable for people with neurological symptoms like spasms, tremors, or spasticity. Additionally, the system detects the force and resistance exerted by the user, which is crucial for control algorithms based on movement intention detection, and provides assistance accordingly. The modular design of the exoskeleton draws inspiration from the Marsi Active Knee (MAK, Marsi Bionics SL, Madrid, Spain) device [23], a certified piece of electro-medical equipment. Adaptations to electric motors and gears were made to provide ample power for assisting flexion and extension movements during various activities. The interconnection of these actuation modules allows for versatility in configurations, accommodating diverse patient characteristics. Each module’s motion is powered by an interchangeable battery located on the torso’s backpack.

The exoskeleton comprises the following components: a trunk structure, two hip modules, two knee modules, two thigh braces, and two calf braces. The trunk structure, featuring a backpack with shoulder straps and an abdominal elastic belt, secures the patient’s trunk and is only necessary when using hip modules. Hip modules can attach to the trunk structure or the corresponding knee module through thigh supports. Knee modules require a calf brace on the structure running parallel to the leg to transmit actuator power.

The actuator, based on the ARES mechanism [22], integrates springs for smooth force transmission, shock absorption, and improved behavior in cases of spasticity. The decentralized architecture employs a 32-bit microcontroller in each module, managing the perception system, communication with other modules, control strategy development, and actuator trajectory commands.

In this architecture, all actuation modules are on the same hierarchical level. Each module’s perception system gathers information on angular position, angular velocity, module orientation with respect to the ground, and actuator torque. 

We employed High Dynamic Force Sensing Resistor Insole technology (IEE Smart Sensing Solutions, Bissen, Luxembourg), equipped with six sensors, distributed as shown in Figure 2. This technology is intricately designed for real-time plantar pressure measurement during locomotion, featuring a triangular cell segmentation that enables precise resistance change detection across a broad pressure range (250 mbar to 7 bar). This specificity is vital for gait analysis research. The durability of these sensors is proven by their capacity to withstand up to 1 million activations in high-humidity conditions with less than 15% lifespan variation. Furthermore, the low hysteresis (<8%) in the pressure response curve accentuates the sensors’ reliability and accuracy in dynamically monitoring pressure changes, a crucial aspect of biomechanical research [24]. Pressure sensors are read at a frequency of 40 Hz and digitized with a 12-bit ADC. The information is sent from the platforms to the knee module on their side via an RS-485 bus and from there to the other modules via a CAN bus. In case there is no knee module on that exoskeleton configuration, the information is sent to the other modules via Bluetooth.

Coordinated trajectory generation for the distributed system relies on an algorithm emulating biological automatisms related to walking—the central pattern generator (CPG) [25]. This algorithm utilizes adaptive frequency oscillators trained to replicate joint angular trajectories and maintain synchronization during movement. The distributed control algorithm allows the computational load to be shared among connected device joints. Further design details are available in [25].

### 2.3. Experimental Protocol

The training regimen for ABI participants comprised three 30 min sessions of robotic gait training. In the initial session, participants universally employed the device in a comprehensive bilateral configuration for evaluation purposes. In subsequent sessions, the module configuration was individually adjusted based on each participant’s specific impairment. These training sessions occurred on non-consecutive days within a one-week timeframe to mitigate the effects of fatigue. The training occurred indoors, within rehabilitation settings, on a smooth and level surface. For safety reasons, a therapist guided the exoskeleton from behind the torso to ensure its stability. 

Healthy participants attended one session with the modular device, during which bilateral configurations (4 modules) and homolateral configurations (2 homolateral modules) were tested.

### 2.4. Outcome Measures

Descriptive information about the participants, including age, weight, and height, was collected. Additionally, the different module configurations used during the study in each session, as well as the average walking time in each session, were recorded.

The plantar pressure was collected at a 40 Hz frequency through sensors integrated into the device itself as participants walked. There are six sensors in total: 1—central metatarsus, 2—Medial calcaneus, 3—lateral calcaneus, 4—medial metatarsus, 5—lateral metatarsus, and 6—hallux. The pressure sensors display information in millibars. Figure 2 shows the location of each sensor within an insole of the right foot.

The MAS [21] was used to assess spasticity in the legs of participants with ABI. This measurement tool classifies resistance to passive movement on a scale from 0 to 4, where 0 indicates no increase in resistance to movement and 4 indicates complete rigidity of the joint in any direction. Spasticity was assessed in both legs during hip and knee flexion and extension movements, as well as during ankle plantar flexion and dorsiflexion.

The muscular strength of participants with ABI was assessed using the Medical Research Council Scale [26] in the sagittal plane movements (both flexion and extension) of the hip, knee, and ankle (plantar flexion and dorsiflexion). Additionally, strength was evaluated in the frontal plane of the hip (both abduction and adduction). This scale categorizes muscle strength on a scale from 0 to 5, where 0 signifies a complete absence of muscle contraction and 5 indicates active movement against gravity with full resistance.

Spasticity and strength assessments were conducted by two physiotherapists specialized in neurorehabilitation and ABI, with years of experience in this field.

### 2.5. Data Processing and Statistical Analysis

The data collected by the device sensors underwent processing to enhance their analytical integrity. This involved the removal of sensor-generated noise and the exclusion of information not gathered during walking. Subsequently, the collected pressure data were converted from mbar to kPa for ease of interpretation. Similarly, the angles recorded for each joint during walking were transformed by dividing them by 100 to further refine the dataset.

We assessed whether the data adhered to the principles of parametric statistics with the Kolmogorov–Smirnov test, alongside Q–Q plots and histograms. Descriptive statistics were utilized to summarize the quantitative data, employing mean and standard deviation (mean ± standard deviation).

The one-way ANOVA test was employed to examine differences in plantar pressures based on the module set used when dealing with three groups of module configurations. The reporting of ANOVA results adhered to American Psychological Association (APA) standards [27], including the F-statistic, significance level, effect size, and statistical power (β-1). This analysis was supplemented with Bonferroni post hoc tests to pinpoint specific group differences [28]. When comparing the healthy and ABI groups, an independent samples t-test was conducted. A significance level of α = 0.05 was set, and differences were considered statistically significant when *p* < 0.05. Furthermore, the Cohen’s d effect size measure was computed to evaluate the magnitude of the effect, applying Cohen’s benchmarks (where d = 0.2 is deemed small, d = 0.5 is considered medium, and d = 0.8 is indicative of a large effect size) [29].

To investigate the impact of lower limb strength, spasticity, and plantar pressures on the module configurations, a multiple linear regression analysis was performed using the forced entry method. Independent variables were incorporated into the regression model, opting for linear regression due to the finite mean, constant variance, and normal distribution of model residuals. The dependent variables were strength and/or spasticity at each joint and movement, while the independent variables included the six plantar sensors, weight, height, and the number of modules used. Due to the extensive amount of data and variables collected in the study (different movement and sides of spasticity and strength measures), a principal component analysis (PCA) was conducted to reduce dimensionality and explore underlying patterns in the data. 

All analyses were performed using RStudio^®^ version 2022.7.2.576 (RStudio, PBC, Boston, MA, USA), IBM^®^ SPSS^®^ Statistics v29 software (IBM Corporation, Armonk, NY, USA), and G*Power Version 3.1.9.6. (Universität Kiel, Kiel, Germany).

## 3. Results

Twenty-one participants (thirteen males and eight females) were recruited for the study, with a mean age of 41.0 ± 13.9 years, an average weight of 71.0 ± 12.7 kg, and a mean height of 170.8 ± 8.5 cm. Participant characteristics and the number of sessions received by module configurations are detailed in Table 1. The average device usage time per session was 21.2 ± 6.9 min.

The means of maximum plantar pressure for each sensor are reflected in Table 2, differentiated by module configuration and participant group. It is observed that the highest peaks of maximum pressure in all configurations are found in sensors 2 and 3 (medial and lateral calcaneus) and sensor 1 (central metatarsus). Higher peaks of maximum pressure are noted in the values of participants with ABI compared to healthy participants across all sensors. An exception is observed in sensor 1, where healthy participants reach the maximum peak.

All pressure values based on the selected module configuration exhibited statistically significant differences at each sensor. Table 3 illustrates the mean pressure differences for module configurations. It is evident that in two-module configurations, higher mean pressure values are obtained in sensors 1, 2, and 3 compared to configurations with four modules. Conversely, four-module configurations resulted in higher pressures in sensors 4 (medial metatarsus), 5 (lateral metatarsus), and 6 (hallux). These differences had a moderate effect (Cohen’s d > 0.4) on sensor 1 and sensor 5. Comparisons involving the three-module configurations should be approached with caution, as they correspond to a single session (Table 1).

Comparisons by participant group can be found in Table 4. All comparisons by sensor and group yielded statistically significant results, with moderate effect sizes (d > 0.4) observed in sensor 3 in the two-module configuration. In the four-module configuration, moderate effect sizes were obtained in sensor 1 (medial metatarsus) and sensor 6. In the four-module configuration, higher plantar pressures (*p* < 0.01) were observed in the ABI group across all sensors. In the two-module configuration, higher plantar pressures were observed in sensors 2, 3, 4, and 6 in the ABI group.

In the principal component analysis, five components were identified, explaining a cumulative variance of 70.09% in the model. These components were identified based on variable loadings as follows: (1) muscle strength (23.59% explained variance), (2) spasticity (19.48%), (3) weight and height (12.73%), (4) metatarsus and hallux sensors (8.38%), and (5) calcaneus sensors (5.55%).

The regression equation used to predict the module configuration in each participant based on the five principal components was statistically significant (F = 28,127.73, *p* < 0.001, β − 1 = 1.0). The R^2^ value was 0.505, indicating that 50% of the variation in the module configuration scores for each participant can be explained by the aforementioned components. The regression coefficients for each component are detailed in Table 5. In the linear regression, data from the first session with the device were not utilized as, per protocol, all participants used the bilateral configuration of four modules, and this could potentially influence the analysis results.

## 4. Discussion

The purpose of this study was to assess the impact of the modular design of a new prototype of an exoskeleton on plantar pressure, both in healthy individuals and those diagnosed with ABI. To date, no study has addressed these evaluations in overground walking exoskeletons, making the values obtained in this work for four-module configurations referential, as there are currently no closer models.

The results revealed that in the three different module configurations (2, 3, and 4), the highest plantar pressure values were found in the hindfoot (medial and lateral calcaneus) in participants with ABI. The sensor with a lower average plantar pressure was located in the hallux, except in the four-module configuration for participants with ABI. In healthy participants, higher values were observed in the medial metatarsus in the two-module configuration. These findings differ from the normalized gait of healthy individuals without an exoskeleton, where higher values are located between the second and third metatarsus, followed by pressure on the hallux [30]. Therefore, it could be interpreted that exoskeleton users do not reach normalized plantar pressures in the midfoot and forefoot regions, being lower in this area during overground walking with an exoskeleton. In this regard, the assistance provided during walking could be an influential factor, since in our study, a therapist guided the device from behind in all sessions (similar to many current devices) to ensure its stability during walking. However, Hayes et al. [31] demonstrated that the ground reaction force during walking in patients using a ReWalk^TM^ (ReWalk Robotics, Inc., Marlborough, MA, USA) with the use of a walker and/or crutches was posteriorized, suggesting that the level of assistance does not seem to influence this finding. Although they did not study plantar pressures during walking, ground reaction forces (forces exerted by the ground on the foot) are directly related to plantar pressures (forces exerted by the foot on the ground). Nevertheless, it might be interesting to assess the influence of the provided assistance on plantar pressures during walking in future studies.

The most significant differences in plantar pressure based on module configuration were observed in the heel and central metatarsus, with higher values in the two-module configuration, while pressures on the hallux and medial and lateral metatarsus were higher in the four-module configuration. This indicates that during the initial contact of the gait cycle, higher pressures are obtained in the heel support of the two-module configuration. However, caution is advised when interpreting values from the three-module configuration, as the number of samples collected in this configuration was significantly lower than in the two- and four-module configurations (Table 1).

When comparing pressures by configuration and group, it was found that in practically all pressure areas, higher values were obtained in the ABI group, with moderate effect sizes in the lateral calcaneus (two modules) and in the central metatarsus and hallux (four modules). Therefore, these three areas may be crucial to better understand variations in plantar pressure in the context of ABI.

The regression model is statistically significant, suggesting that the module configuration can be predicted to some extent based on the predictor variables included in the model. Each variable makes a unique contribution to predicting the module configuration. A positive correlation is evident between pressure on the front sensors and the number of modules used, whereas plantar pressures in the hindfoot show a negative correlation with the number of modules utilized. A plausible hypothesis could be that patients with equine foot tend to have increased pressure in the front, while those exhibiting proper heel support may display higher functional walking levels compared to the former. The regression analysis used variables from the PCA to reduce multicollinearity among all variables related to force and spasticity movements, enhancing the reliability of the model. However, synthesizing all force and spasticity movements into a single component might have reduced precision in the analysis. Nevertheless, the absolute magnitudes of the coefficients indicate that weight, height, and spasticity had the most significant influence on the model.

Finally, these changes in plantar pressure due to module configurations may introduce variability into the gait cycle rehabilitation process, influencing it based on each patient’s context. Motor learning principles, encompassing task specificity, intensity, and variability, are crucial in this context [32]. Task variability during rehabilitation is considered beneficial, as it can contribute to the learning and relearning of tasks in patients with ABI [33]. Introducing changes and adaptations in plantar pressure through different module configurations could enhance neuronal plasticity and promote motor system adaptation, thereby benefiting the rehabilitation and gait improvement process in individuals with ABI [34]. Therefore, we can make a cautious reading and indicate that it is important to manage the dosage of the treatment well and include variability with the use of the modules.

This study has several limitations. Firstly, there were variations in the data collected between module configurations and across participant groups, including differences in participant numbers for each configuration. Secondly, we acknowledge a limitation in our methodology: the sample size was not statistically predetermined, as this study was intended as a pilot or proof-of-concept for future research. Additionally, the inclusion criteria addressed an age eligibility range of 18 to 85 years, as the study focused on the adult population, thus encompassing a broad age range that could introduce additional variability into the results. However, the participants ultimately included in the study had an average age of 41.0 ± 13.9, reflecting a narrower range of variability than the inclusion criteria allowed. This demonstrates that, despite the wide age range criteria, the actual age distribution of participants did not adversely affect the study’s management or outcomes.

Despite these challenges, the gathered results provide a foundational framework for more rigorous future studies. It is advised that upcoming research focuses on analyzing plantar pressure in devices of this nature during walking to facilitate meaningful comparisons between different devices. This approach will contribute to a deeper understanding of the specific effects of varying exoskeleton configurations on plantar pressure, enabling more comprehensive insights into their rehabilitative potential. Finally, leveraging the modular design of such exoskeletons, it could be studied how the rest of the possible module configurations (Figure 1) affect these plantar pressures. During the present study, only three configurations were used due to the motor impairment of the users.

## 5. Conclusions

Modularity design of overground exoskeletons exerts a significant impact on the distribution of plantar pressure in users. Configurations with two modules were associated with elevated pressures in the heel and central metatarsus, contrasting with configurations with four modules, which exhibited higher pressures in the metatarsus and hallux. Additionally, participants with ABI experienced higher plantar pressures during walking compared to healthy participants.

## Figures and Tables

**Figure 1 sensors-24-01435-f001:**
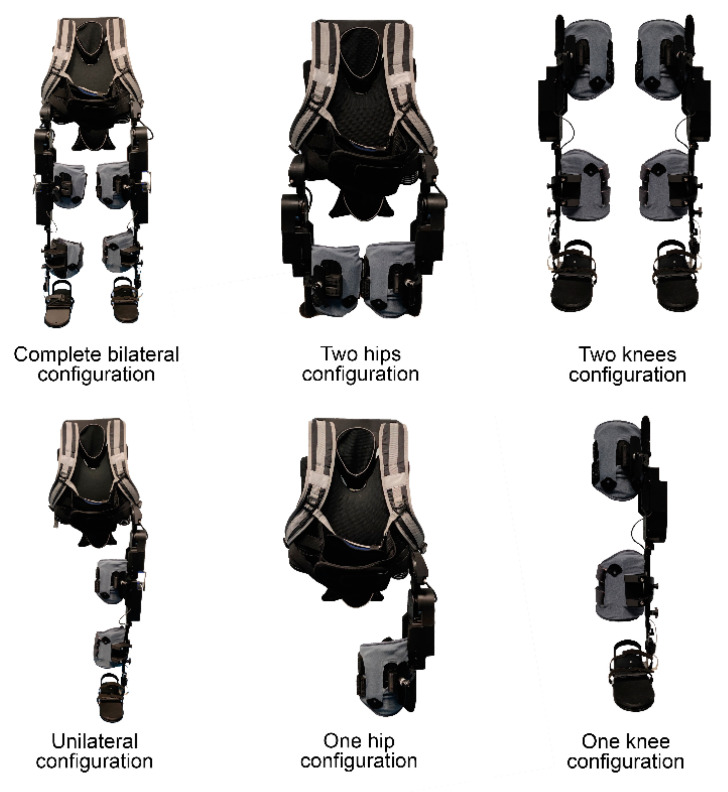
Six different module configurations of STELO.

**Figure 2 sensors-24-01435-f002:**
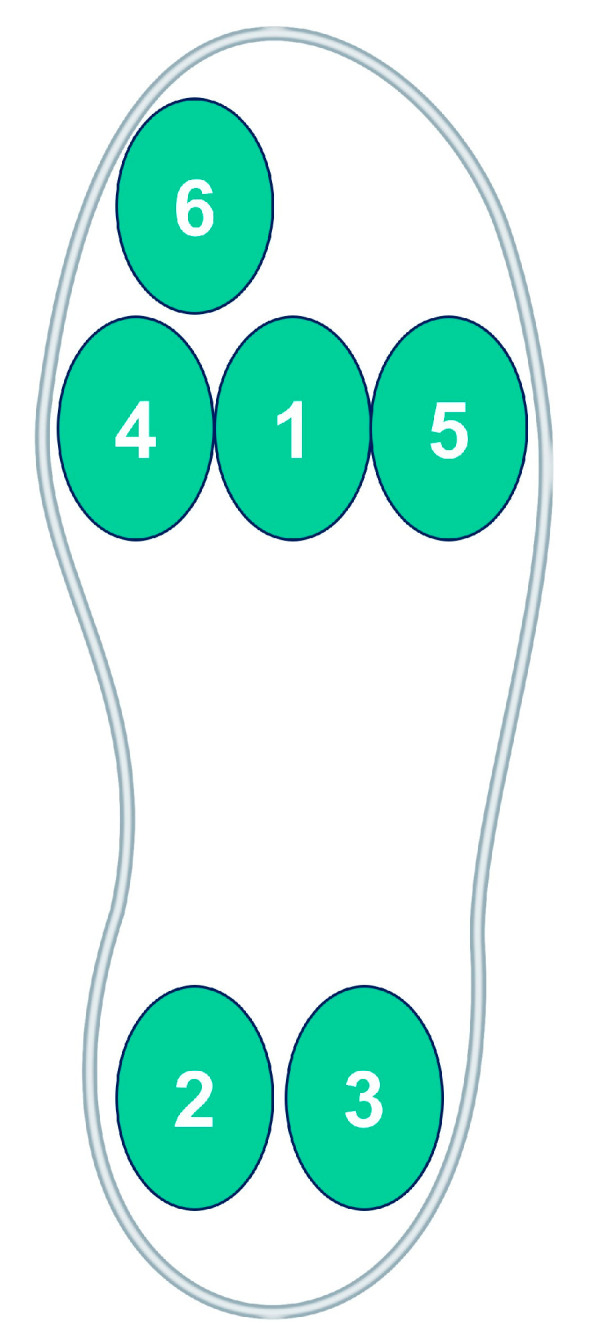
Location of each sensor within an insole of the right foot. 1—central metatarsus, 2—Medial calcaneus, 3—lateral calcaneus, 4—medial metatarsus, 5—lateral metatarsus, and 6—hallux.

**Table 1 sensors-24-01435-t001:** Description of the sample.

Group	n	M/F	Age (Years)	Weight (kg)	Height (cm)	Four Modules (n Sessions)	Two Modules (n Sessions)	Three Modules (n Sessions)
ABI	14	8/7	42.0 ± 10.8	70.2 ± 14.9	170.2 ± 9.0	26	15	1
Healthy	7	5/2	40.1 ± 11.7	72.4 ± 6.8	172.0 ± 8.1	7 ^a^	7 ^a^	

ABI—Acquired Brain Injury. M—male. F—female. Four modules—two hips and two knees. Two modules—one hip and one knee of same side (homolateral). Three modules—two hips and one knee. ^a^ During the 7 sessions of healthy participants, both reflected configurations (four and two modules) were utilized within the same session, resulting in a total of 7 uses per configuration.

**Table 2 sensors-24-01435-t002:** Mean peak plantar pressure (kPa) results including all groups’ means ± standard deviation.

Mean Peak Pressure (kPa) by Module Configuration and Group					
Sensors	Two Modules		Three Modules	Four Modules	
	Healthy	ABI	ABI	Healthy	ABI
1	677.71 ± 126.36	583.07 ± 153.44	625.21 ± 139.08	290.47 ± 151.73	467.32 ± 232.89
2	514.85 ± 27.48	759.66 ± 154.06	492.35 ± 88.22	491.15 ± 50.66	848.57 ± 168.20
3	562.85 ± 117.55	793.54 ± 181.87	608.71 ± 87.65	494.14 ± 54.44	799.96 ± 175.16
4	260.71 ± 127.82	273.19 ± 93.36	330.75 ± 66.64	181.57 ± 57.10	249.52 ± 135.53
5	402.33 ± 77.24	366.05 ± 66.98	391.51 ± 56.39	271.57 ± 137.94	408.44 ± 32.49
6	229.81 ± 86.77	332.07 ± 119.49	345.20 ± 58.72	186.97 ± 53.21	414.56 ± 210.75

ABI—Acquired Brain Injury. Sensors: 1—central metatarsus, 2—Medial calcaneus, 3—lateral calcaneus, 4—medial metatarsus, 5—lateral metatarsus, and 6—hallux.

**Table 3 sensors-24-01435-t003:** Mean differences per module configuration in plantar pressure including confidence interval and effect size.

	Two Modules–Three Modules		Two Modules–Four Modules		Three Modules–Four Modules	
Sensors	Mean Differences (CI)	ES	Mean Differences (CI)	ES	Mean Differences (CI)	ES
1	−31.53 (−33.33 to −29.73) *	0.19	46.63 (43.00 to 44.27) *	0.41	75.16 (73.41 to 76.91) *	0.53
2	126.95 (123.85 to 130.06) *	0.44	82.01 (80.90 to 83.11) *	0.27	−44.94 (−47.97 to −41.92) *	0.17
3	95.83 (92.82 to 98.83) *	0.64	40.38 (39.31 to 41.45) *	0.36	−55.45 (−58.38 to −52.52) *	0.35
4	−8.70 (−9.54 to −7.86) *	0.20	−12.34 (−12.64 to −12.04) *	0.10	−3.64 (−4.47 to −2.82) *	0.29
5	5.41 (4.23 to 6.57) *	0.23	−29.40 (−29.82 to −28.98) *	0.42	−34.81 (−35.95 to −33.67) *	0.61
6	−6.99 (−7.96 to −6.03) *	0.28	−14.97 (−15.31 to 14.63) *	0.18	−7.98 (−8.91 to −7.04) *	0.45

CI—95% confidence interval. ES—effect size (Cohen’s d). *—significant difference between groups, *p* < 0.01. Sensors: 1—central metatarsus, 2—Medial calcaneus, 3—lateral calcaneus, 4—medial metatarsus, 5—lateral metatarsus, and 6—hallux.

**Table 4 sensors-24-01435-t004:** Mean differences per group and module configuration in plantar pressure including confidence interval and effect size.

	Two Modules		Four Modules	
Sensors	Mean Differences (CI)	ES	Mean Differences (CI)	ES
1	33.70 (32–37 to 35.02) *	0.23	−35.21 (−35.66 to −34.77) *	0.44
2	−72.78 (−75.08 to −70.48) *	0.31	−7.21 (−8.03 to −6.39) *	0.04
3	−97.85 (−99.83 to −95.88) *	0.48	−10.50 (−11.38 to −9.62)	0.06
4	−2.58 (−3.14 to −2.03) *	0.04	−10.45 (−10.68 to −10.21) *	0.24
5	13.97 (13.40 to 14.54) *	0.24	−27.37 (−27.76 to −26.97) *	0.39
6	−8.83 (−9.25 to −8.42) *	0.15	−25.10 (−25.38 to −24.83) *	0.49

CI—95% confidence interval. ES—effect size (Cohen’s d). ABI—Acquired Brain Injury. *—significant difference between groups, *p* < 0.01. Sensors: 1—central metatarsus, 2—Medial calcaneus, 3—lateral calcaneus, 4—medial metatarsus, 5—lateral metatarsus, and 6—hallux. Note: group differences between the three-module configurations are not reflected, as this configuration was only used by participants with ABI.

**Table 5 sensors-24-01435-t005:** Linear regression model regarding module configuration.

	Coefficient	*p* Value	Confidence Interval 95%		Collinearity	
			Inferior	Superior	Tolerance	VIF
Constant	2.892	<0.001	2.887	2.896		
1—strength	0.088	<0.001	0.083	0.093	0.524	1.907
2—spasticity	−0.488	<0.001	−0.497	−0.48	0.641	1.559
3—weight and height	0.585	<0.001	0.580	0.590	0.691	1.447
4—metatarsus and hallux pressure	0.349	<0.001	0.345	0.354	0.644	1.554
5—calcaneal pressure	−0.245	<0.001	−0.248	−0.242	0.918	1.089
R^2^	0.505					
R^2^ adj	0.505					

VIF—Variance Inflation Factor.

## Data Availability

Marsi Bionics SL is committed to providing access to de-identified patient-level data that underpin the findings presented in this article, in response to scientifically valid research proposals. Qualified researchers are encouraged to submit requests for access to these data, which will be thoughtfully considered by Marsi Bionics SL. While every effort will be made to accommodate legitimate research requests, it is important to acknowledge that certain circumstances may impede the retrieval or delivery of data. These factors may include considerations related to patient privacy, necessary permissions, contractual obligations, and potential conflicts of interest. All individuals granted access to the data will be required to enter into a data use agreement provided by Marsi Bionics SL. This agreement will outline the terms and conditions governing the use of the data, ensuring responsible and ethical handling in accordance with applicable regulations and standards. Marsi Bionics SL remains committed to facilitating access to data for the advancement of scientific knowledge while upholding ethical and legal considerations that safeguard patient privacy and maintain the integrity of research.

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
