# Peer review of "Modularity Implications of an Overground Exoskeleton on Plantar Pressures, Strength, and Spasticity in Persons with Acquired Brain Injury"

_sensors, 2024, doi:10.3390/s24051435_

Round 1

Reviewer 1 Report

Comments and Suggestions for Authors

Dear Authors,

Kindly address the following:

1. Whether including such broad range  age within the range of 18 to 85 years with ABI will not effect the management and results? Kindly explain about this in Introduction or in Discussion.

2. Need to explain about the calcaneal and other pressures in detail and its impact?

3. Did you observed any differences between male vs female participants? If so, explain the reasons.

4. Reliability and validity of using this equipment is much needed for this kind of study.

Author Response

Response to the comments of
Manuscript ID sensors 2876754
Thank you for your kind comments and revisions to the manuscript, which have helped us to
improve the quality of the manuscript and improve some aspects. In the following section we
show your comments, and our response and/or correction marked. In addition, we attach the
tracked changes version of the manuscript as journal guidelines suggested.
We appreciate your engagement with our work and are open to any further questions or
suggestions you may have.
The
a uthors
Reviewer 1
Dear Authors, Kindly address the following:
1. Whether including such broad range age within the range of 18 to 85 years with ABI will not effect the management and results? Kindly explain about this in Introduction or in Discussion. The inclusion criteria addressed an age eligibility range of 18 to 85 years, as the study focused on the adult population, thus encompassing a broad age range that could introduce additional variability into the results. However, the participants ultimately included in the study had an average age of 41.0 ± 13.9, reflecting a narrower range of variability than the inclusion criteria allowed. This demonstrates that, despite the wide age range criteria, the actual age distribution of participants did not adversely affect the study's management or outcomes. This statement is included in study limitations paragraph (discussion).
2. Need to explain about the calcaneal and other pressures in detail and its impact? It has been included in introduction: Plantar pressures in various regions of the foot have significant implications for individuals with ABI, affecting mobility and the risk of foot-related complications. High plantar pressures, especially in the central metatarsus, hallux, and medial metatarsus, are associated with overuse injuries in these areas, highlighting the need for targeted interventions to prevent or manage such injuries in patients with ABI (15) The use of high-heeled footwear shifts pressure and stress from the heel to the metatarsus and hallux, increasing the risk of deformities and pain, suggesting that footwear choice is crucial for managing foot health in ABI patients (16). Additionally, biomechanical alterations, such as those caused by hallux valgus, modify plantar pressure distribution, increasing the load on the lateral metatarsus while reducing pressure under the hallux, potentially contributing to the development of metatarsalgia (17). These changes in plantar pressure distribution may have significant clinical implications, including the development of pain, deformities, and decreased foot functionality, emphasizing the importance of preventive and therapeutic interventions aimed at improving plantar pressure distribution in at-risk individuals, such as those with ABI.
3. Did you observed any differences between male vs female participants? If so, explain the reasons. Gender-based analyses were not conducted as they were not a focus of the study, which aimed to explore the novel aspects of modularity and differences between ABI patients and healthy individuals. Furthermore, gender analyses were omitted since existing literature has already documented differences between genders (Putti, Arnold, & Abboud, 2010).
2
4. Reliability and validity of using this equipment is much needed for this kind of study. It has been added in methods section:
We employed High Dynamic Force Sensing Resistor Insole technology (IEE Smart Sensing Solutions, Bissen, Luxembourg), equipped with six sensors distributed as shown in Figure 2. This technology is intricately designed for real time plantar pressure measurement during locomotion, featuring a triangular cell segmentation that enables precise resistance change detection across a broad pressure range (250 mbar to 7 bar). This specificity is vital for gait analysis research. The durability of these sensors is proven by their capacity to withstand up to 1 million activations in high humidity conditions with less than 15% lifespan variation. Furthermore, the low hysteresis (<8%) in the pressure response curve accentuates the sensors' reliability and accuracy in dynamically monitoring pressure changes, a crucial aspect of biomechanical research (24). Pressure sensors are read at a frequency of 40 Hz and digitized with a 12-bit ADC. The information is sent from the platforms to the knee module on their side via an RS-485 bus and from there to the other modules via a CAN bus. In case there is no knee module on that exoskeleton configuration, the information is sent to the other modules via Bluetooth.
Reviewer 2
Reviewer 2
The manuscript presents a study on the effects of a modular overground exoskeleton on plantar pressure
The manuscript presents a study on the effects of a modular overground exoskeleton on plantar pressure distribution among individuals with acquired brain injury (ABI) and healthy controls. The study explores distribution among individuals with acquired brain injury (ABI) and healthy controls. The study explores how different configurations of the exoskeleton how different configurations of the exoskeleton impact plantar pressures, muscle strength, and spasticity. impact plantar pressures, muscle strength, and spasticity. The findings suggest significant differences in plantar pressures between the ABI and control groups, The findings suggest significant differences in plantar pressures between the ABI and control groups, influenced by the exoskeleton's modularity. It highlights the potential of customized rehabilitinfluenced by the exoskeleton's modularity. It highlights the potential of customized rehabilitation ation technologies to enhance functional recovery in ABI patients.technologies to enhance functional recovery in ABI patients.
We appreciate your recognition of the manuscript's contribution to understanding the effects of a modular
We appreciate your recognition of the manuscript's contribution to understanding the effects of a modular overground exoskeleton on plantar pressure distribution among individuals with ABI and healthy controls.overground exoskeleton on plantar pressure distribution among individuals with ABI and healthy controls.
Reviewer 3
Reviewer 3
Thank you very much for letting me review this very interesting article, “Modularity implications of an
Thank you very much for letting me review this very interesting article, “Modularity implications of an overground exoskeleton on plantar pressures, strength, and spasticity in persons with acquired brain injury”. overground exoskeleton on plantar pressures, strength, and spasticity in persons with acquired brain injury”. The study is interesting and novel, and it seems a promising to meet the needs of acquired brain injury The study is interesting and novel, and it seems a promising to meet the needs of acquired brain injury patients. The article is generally well wrpatients. The article is generally well written.itten.
Around 21 participants were included in the study and the results show significant differences in plantar
Around 21 participants were included in the study and the results show significant differences in plantar pressure among participants, which is strongly influenced by the exoskeleton’s modularity. Since there is pressure among participants, which is strongly influenced by the exoskeleton’s modularity. Since there is notable significant difference between ABI pnotable significant difference between ABI patients and healthy individuals which suggests STELO can be atients and healthy individuals which suggests STELO can be valuable device. But the main concern to claim STELO as valuable device is the sample size of patients valuable device. But the main concern to claim STELO as valuable device is the sample size of patients used in this study is too low.used in this study is too low.
Also, did the author calculate the sample size before performing their assessments? It is not mentioned in
Also, did the author calculate the sample size before performing their assessments? It is not mentioned in the methods section.the methods section.
Thank you very much for your thoughtful review and positive remarks on our article, "Modularity
Thank you very much for your thoughtful review and positive remarks on our article, "Modularity implications of an overground exoskeleton on plantar pressures, strength, and spasticity in persons with implications of an overground exoskeleton on plantar pressures, strength, and spasticity in persons with acquired brain injury." We are grateful for your recognacquired brain injury." We are grateful for your recognition of the study's novelty and its potential ition of the study's novelty and its potential contribution to the field.contribution to the field.
Regarding the concerns about the sample size, we acknowledge that the study involved 21 participants,
Regarding the concerns about the sample size, we acknowledge that the study involved 21 participants, which might be considered limited for generalizing the findings. The sample size was determined based on which might be considered limited for generalizing the findings. The sample size was determined based on preliminary data and logistical constraints, aimipreliminary data and logistical constraints, aiming to balance between a rigorous methodological approach ng to balance between a rigorous methodological approach and the feasibility of conducting specialized research within this patient population. However, we recognize and the feasibility of conducting specialized research within this patient population. However, we recognize the importance of a larger sample size for enhancing the study's robustness and will consithe importance of a larger sample size for enhancing the study's robustness and will consider this in future der this in future research.research.
3
It has been included in limitation section:
It has been included in limitation section: “We acknowledge a limitation in our methodology: the sample “We acknowledge a limitation in our methodology: the sample size was not statistically predetermined, as this study was intended as a pilot or proofsize was not statistically predetermined, as this study was intended as a pilot or proof--ofof--concept for future concept for future research”.research”.

Reviewer 2 Report

Comments and Suggestions for Authors

The manuscript presents a study on the effects of a modular overground exoskeleton on plantar pressure distribution among individuals with acquired brain injury (ABI) and healthy controls. The study explores how different configurations of the exoskeleton impact plantar pressures, muscle strength, and spasticity. The findings suggest significant differences in plantar pressures between the ABI and control groups, influenced by the exoskeleton's modularity. It highlights the potential of customized rehabilitation technologies to enhance functional recovery in ABI patients.

Author Response

Response to the comments of
M anuscript ID sensors 2876754
Thank you for your kind comments and revisions to the manuscript, which have helped us to
improve the quality of the manuscript and improve some aspects. In the following section we
show your comments, and our response and/or correction marked. In addition, we attach the
tracked changes version of the manuscript as journal guidelines suggested.
We appreciate your engagement with our work and are open to any further questions or
suggestions you may have.
The
a uthors
Reviewer 1
Dear Authors, Kindly address the following:
1. Whether including such broad range age within the range of 18 to 85 years with ABI will not effect the management and results? Kindly explain about this in Introduction or in Discussion. The inclusion criteria addressed an age eligibility range of 18 to 85 years, as the study focused on the adult population, thus encompassing a broad age range that could introduce additional variability into the results. However, the participants ultimately included in the study had an average age of 41.0 ± 13.9, reflecting a narrower range of variability than the inclusion criteria allowed. This demonstrates that, despite the wide age range criteria, the actual age distribution of participants did not adversely affect the study's management or outcomes. This statement is included in study limitations paragraph (discussion).
2. Need to explain about the calcaneal and other pressures in detail and its impact? It has been included in introduction: Plantar pressures in various regions of the foot have significant implications for individuals with ABI, affecting mobility and the risk of foot-related complications. High plantar pressures, especially in the central metatarsus, hallux, and medial metatarsus, are associated with overuse injuries in these areas, highlighting the need for targeted interventions to prevent or manage such injuries in patients with ABI (15) The use of high-heeled footwear shifts pressure and stress from the heel to the metatarsus and hallux, increasing the risk of deformities and pain, suggesting that footwear choice is crucial for managing foot health in ABI patients (16). Additionally, biomechanical alterations, such as those caused by hallux valgus, modify plantar pressure distribution, increasing the load on the lateral metatarsus while reducing pressure under the hallux, potentially contributing to the development of metatarsalgia (17). These changes in plantar pressure distribution may have significant clinical implications, including the development of pain, deformities, and decreased foot functionality, emphasizing the importance of preventive and therapeutic interventions aimed at improving plantar pressure distribution in at-risk individuals, such as those with ABI.
3. Did you observed any differences between male vs female participants? If so, explain the reasons. Gender-based analyses were not conducted as they were not a focus of the study, which aimed to explore the novel aspects of modularity and differences between ABI patients and healthy individuals. Furthermore, gender analyses were omitted since existing literature has already documented differences between genders (Putti, Arnold, & Abboud, 2010).
2
4. Reliability and validity of using this equipment is much needed for this kind of study. It has been added in methods section:
We employed High Dynamic Force Sensing Resistor Insole technology (IEE Smart Sensing Solutions, Bissen, Luxembourg), equipped with six sensors distributed as shown in Figure 2. This technology is intricately designed for real time plantar pressure measurement during locomotion, featuring a triangular cell segmentation that enables precise resistance change detection across a broad pressure range (250 mbar to 7 bar). This specificity is vital for gait analysis research. The durability of these sensors is proven by their capacity to withstand up to 1 million activations in high humidity conditions with less than 15% lifespan variation. Furthermore, the low hysteresis (<8%) in the pressure response curve accentuates the sensors' reliability and accuracy in dynamically monitoring pressure changes, a crucial aspect of biomechanical research (24). Pressure sensors are read at a frequency of 40 Hz and digitized with a 12-bit ADC. The information is sent from the platforms to the knee module on their side via an RS-485 bus and from there to the other modules via a CAN bus. In case there is no knee module on that exoskeleton configuration, the information is sent to the other modules via Bluetooth.
Reviewer 2
Reviewer 2
The manuscript presents a study on the effects of a modular overground exoskeleton on plantar pressure
The manuscript presents a study on the effects of a modular overground exoskeleton on plantar pressure distribution among individuals with acquired brain injury (ABI) and healthy controls. The study explores distribution among individuals with acquired brain injury (ABI) and healthy controls. The study explores how different configurations of the exoskeleton how different configurations of the exoskeleton impact plantar pressures, muscle strength, and spasticity. impact plantar pressures, muscle strength, and spasticity. The findings suggest significant differences in plantar pressures between the ABI and control groups, The findings suggest significant differences in plantar pressures between the ABI and control groups, influenced by the exoskeleton's modularity. It highlights the potential of customized rehabilitinfluenced by the exoskeleton's modularity. It highlights the potential of customized rehabilitation ation technologies to enhance functional recovery in ABI patients.technologies to enhance functional recovery in ABI patients.
We appreciate your recognition of the manuscript's contribution to understanding the effects of a modular
We appreciate your recognition of the manuscript's contribution to understanding the effects of a modular overground exoskeleton on plantar pressure distribution among individuals with ABI and healthy controls.overground exoskeleton on plantar pressure distribution among individuals with ABI and healthy controls.
Reviewer 3
Reviewer 3
Thank you very much for letting me review this very interesting article, “Modularity implications of an
Thank you very much for letting me review this very interesting article, “Modularity implications of an overground exoskeleton on plantar pressures, strength, and spasticity in persons with acquired brain injury”. overground exoskeleton on plantar pressures, strength, and spasticity in persons with acquired brain injury”. The study is interesting and novel, and it seems a promising to meet the needs of acquired brain injury The study is interesting and novel, and it seems a promising to meet the needs of acquired brain injury patients. The article is generally well wrpatients. The article is generally well written.itten.
Around 21 participants were included in the study and the results show significant differences in plantar
Around 21 participants were included in the study and the results show significant differences in plantar pressure among participants, which is strongly influenced by the exoskeleton’s modularity. Since there is pressure among participants, which is strongly influenced by the exoskeleton’s modularity. Since there is notable significant difference between ABI pnotable significant difference between ABI patients and healthy individuals which suggests STELO can be atients and healthy individuals which suggests STELO can be valuable device. But the main concern to claim STELO as valuable device is the sample size of patients valuable device. But the main concern to claim STELO as valuable device is the sample size of patients used in this study is too low.used in this study is too low.
Also, did the author calculate the sample size before performing their assessments? It is not mentioned in
Also, did the author calculate the sample size before performing their assessments? It is not mentioned in the methods section.the methods section.
Thank you very much for your thoughtful review and positive remarks on our article, "Modularity
Thank you very much for your thoughtful review and positive remarks on our article, "Modularity implications of an overground exoskeleton on plantar pressures, strength, and spasticity in persons with implications of an overground exoskeleton on plantar pressures, strength, and spasticity in persons with acquired brain injury." We are grateful for your recognacquired brain injury." We are grateful for your recognition of the study's novelty and its potential ition of the study's novelty and its potential contribution to the field.contribution to the field.
Regarding the concerns about the sample size, we acknowledge that the study involved 21 participants,
Regarding the concerns about the sample size, we acknowledge that the study involved 21 participants, which might be considered limited for generalizing the findings. The sample size was determined based on which might be considered limited for generalizing the findings. The sample size was determined based on preliminary data and logistical constraints, aimipreliminary data and logistical constraints, aiming to balance between a rigorous methodological approach ng to balance between a rigorous methodological approach and the feasibility of conducting specialized research within this patient population. However, we recognize and the feasibility of conducting specialized research within this patient population. However, we recognize the importance of a larger sample size for enhancing the study's robustness and will consithe importance of a larger sample size for enhancing the study's robustness and will consider this in future der this in future research.research.
3
It has been included in limitation section:
It has been included in limitation section: “We acknowledge a limitation in our methodology: the sample “We acknowledge a limitation in our methodology: the sample size was not statistically predetermined, as this study was intended as a pilot or proofsize was not statistically predetermined, as this study was intended as a pilot or proof--ofof--concept for future concept for future research”

Reviewer 3 Report

Comments and Suggestions for Authors

Thank you very much for letting me review this very interesting article, “Modularity implications of an overground exoskeleton on plantar pressures, strength, and spasticity in persons with acquired brain injury”. The study is interesting and novel, and it seems a promising to meet the needs of acquired brain injury patients. The article is generally well written.

Around 21 participants were included in the study and the results show significant differences in plantar pressure among participants, which is strongly influenced by the exoskeleton’s modularity. Since there is notable significant difference between ABI patients and healthy individuals which suggests STELO can be valuable device. But the main concern to claim STELO as valuable device is the sample size of patients used in this study is too low.

Also, did the author calculate the sample size before performing their assessments? It is not mentioned in the methods section.

Author Response

Response to the comments of
M anuscript ID sensors 2876754
Thank you for your kind comments and revisions to the manuscript, which have helped us to
improve the quality of the manuscript and improve some aspects. In the following section we
show your comments, and our response and/or correction marked. In addition, we attach the
tracked changes version of the manuscript as journal guidelines suggested.
We appreciate your engagement with our work and are open to any further questions or
suggestions you may have.
The
a uthors
Reviewer 1
Dear Authors, Kindly address the following:
1. Whether including such broad range age within the range of 18 to 85 years with ABI will not effect the management and results? Kindly explain about this in Introduction or in Discussion. The inclusion criteria addressed an age eligibility range of 18 to 85 years, as the study focused on the adult population, thus encompassing a broad age range that could introduce additional variability into the results. However, the participants ultimately included in the study had an average age of 41.0 ± 13.9, reflecting a narrower range of variability than the inclusion criteria allowed. This demonstrates that, despite the wide age range criteria, the actual age distribution of participants did not adversely affect the study's management or outcomes. This statement is included in study limitations paragraph (discussion).
2. Need to explain about the calcaneal and other pressures in detail and its impact? It has been included in introduction: Plantar pressures in various regions of the foot have significant implications for individuals with ABI, affecting mobility and the risk of foot-related complications. High plantar pressures, especially in the central metatarsus, hallux, and medial metatarsus, are associated with overuse injuries in these areas, highlighting the need for targeted interventions to prevent or manage such injuries in patients with ABI (15) The use of high-heeled footwear shifts pressure and stress from the heel to the metatarsus and hallux, increasing the risk of deformities and pain, suggesting that footwear choice is crucial for managing foot health in ABI patients (16). Additionally, biomechanical alterations, such as those caused by hallux valgus, modify plantar pressure distribution, increasing the load on the lateral metatarsus while reducing pressure under the hallux, potentially contributing to the development of metatarsalgia (17). These changes in plantar pressure distribution may have significant clinical implications, including the development of pain, deformities, and decreased foot functionality, emphasizing the importance of preventive and therapeutic interventions aimed at improving plantar pressure distribution in at-risk individuals, such as those with ABI.
3. Did you observed any differences between male vs female participants? If so, explain the reasons. Gender-based analyses were not conducted as they were not a focus of the study, which aimed to explore the novel aspects of modularity and differences between ABI patients and healthy individuals. Furthermore, gender analyses were omitted since existing literature has already documented differences between genders (Putti, Arnold, & Abboud, 2010).
2
4. Reliability and validity of using this equipment is much needed for this kind of study. It has been added in methods section:
We employed High Dynamic Force Sensing Resistor Insole technology (IEE Smart Sensing Solutions, Bissen, Luxembourg), equipped with six sensors distributed as shown in Figure 2. This technology is intricately designed for real time plantar pressure measurement during locomotion, featuring a triangular cell segmentation that enables precise resistance change detection across a broad pressure range (250 mbar to 7 bar). This specificity is vital for gait analysis research. The durability of these sensors is proven by their capacity to withstand up to 1 million activations in high humidity conditions with less than 15% lifespan variation. Furthermore, the low hysteresis (<8%) in the pressure response curve accentuates the sensors' reliability and accuracy in dynamically monitoring pressure changes, a crucial aspect of biomechanical research (24). Pressure sensors are read at a frequency of 40 Hz and digitized with a 12-bit ADC. The information is sent from the platforms to the knee module on their side via an RS-485 bus and from there to the other modules via a CAN bus. In case there is no knee module on that exoskeleton configuration, the information is sent to the other modules via Bluetooth.
Reviewer 2
Reviewer 2
The manuscript presents a study on the effects of a modular overground exoskeleton on plantar pressure
The manuscript presents a study on the effects of a modular overground exoskeleton on plantar pressure distribution among individuals with acquired brain injury (ABI) and healthy controls. The study explores distribution among individuals with acquired brain injury (ABI) and healthy controls. The study explores how different configurations of the exoskeleton how different configurations of the exoskeleton impact plantar pressures, muscle strength, and spasticity. impact plantar pressures, muscle strength, and spasticity. The findings suggest significant differences in plantar pressures between the ABI and control groups, The findings suggest significant differences in plantar pressures between the ABI and control groups, influenced by the exoskeleton's modularity. It highlights the potential of customized rehabilitinfluenced by the exoskeleton's modularity. It highlights the potential of customized rehabilitation ation technologies to enhance functional recovery in ABI patients.technologies to enhance functional recovery in ABI patients.
We appreciate your recognition of the manuscript's contribution to understanding the effects of a modular
We appreciate your recognition of the manuscript's contribution to understanding the effects of a modular overground exoskeleton on plantar pressure distribution among individuals with ABI and healthy controls.overground exoskeleton on plantar pressure distribution among individuals with ABI and healthy controls.
Reviewer 3
Reviewer 3
Thank you very much for letting me review this very interesting article, “Modularity implications of an
Thank you very much for letting me review this very interesting article, “Modularity implications of an overground exoskeleton on plantar pressures, strength, and spasticity in persons with acquired brain injury”. overground exoskeleton on plantar pressures, strength, and spasticity in persons with acquired brain injury”. The study is interesting and novel, and it seems a promising to meet the needs of acquired brain injury The study is interesting and novel, and it seems a promising to meet the needs of acquired brain injury patients. The article is generally well wrpatients. The article is generally well written.itten.
Around 21 participants were included in the study and the results show significant differences in plantar
Around 21 participants were included in the study and the results show significant differences in plantar pressure among participants, which is strongly influenced by the exoskeleton’s modularity. Since there is pressure among participants, which is strongly influenced by the exoskeleton’s modularity. Since there is notable significant difference between ABI pnotable significant difference between ABI patients and healthy individuals which suggests STELO can be atients and healthy individuals which suggests STELO can be valuable device. But the main concern to claim STELO as valuable device is the sample size of patients valuable device. But the main concern to claim STELO as valuable device is the sample size of patients used in this study is too low.used in this study is too low.
Also, did the author calculate the sample size before performing their assessments? It is not mentioned in
Also, did the author calculate the sample size before performing their assessments? It is not mentioned in the methods section.the methods section.
Thank you very much for your thoughtful review and positive remarks on our article, "Modularity
Thank you very much for your thoughtful review and positive remarks on our article, "Modularity implications of an overground exoskeleton on plantar pressures, strength, and spasticity in persons with implications of an overground exoskeleton on plantar pressures, strength, and spasticity in persons with acquired brain injury." We are grateful for your recognacquired brain injury." We are grateful for your recognition of the study's novelty and its potential ition of the study's novelty and its potential contribution to the field.contribution to the field.
Regarding the concerns about the sample size, we acknowledge that the study involved 21 participants,
Regarding the concerns about the sample size, we acknowledge that the study involved 21 participants, which might be considered limited for generalizing the findings. The sample size was determined based on which might be considered limited for generalizing the findings. The sample size was determined based on preliminary data and logistical constraints, aimipreliminary data and logistical constraints, aiming to balance between a rigorous methodological approach ng to balance between a rigorous methodological approach and the feasibility of conducting specialized research within this patient population. However, we recognize and the feasibility of conducting specialized research within this patient population. However, we recognize the importance of a larger sample size for enhancing the study's robustness and will consithe importance of a larger sample size for enhancing the study's robustness and will consider this in future der this in future research.research.
3
It has been included in limitation section:
It has been included in limitation section: “We acknowledge a limitation in our methodology: the sample “We acknowledge a limitation in our methodology: the sample size was not statistically predetermined, as this study was intended as a pilot or proofsize was not statistically predetermined, as this study was intended as a pilot or proof--ofof--concept for future concept for future research”.research”.
